# Challenges in Preparation of Albumin Nanoparticle-Based Radiopharmaceuticals

**DOI:** 10.3390/molecules27238596

**Published:** 2022-12-06

**Authors:** James R. Ballinger

**Affiliations:** School of Biomedical Engineering and Imaging Sciences, King’s College London, London WC2R 2LS, UK; jim.ballinger@kcl.ac.uk

**Keywords:** human serum albumin, nanocolloid, sentinel lymph node, technetium-99m, gallium-68, zirconium-89, indocyanine green

## Abstract

Albumin nanocolloids have been used as radiopharmaceuticals for more than 40 years. Their main use is in lymphoscintigraphy and the detection of the sentinel lymph node as part of the surgical treatment of a variety of solid tumours. The main licensed products are labelled with the gamma emitter technetium-99m. Recently, two analogues labelled with positron emitters have been reported, using gallium-68 and zirconium-89. For about 10 years, there has been interest in dual-modal agents with both radioactive and fluorescent labels to improve the localisation of the sentinel lymph node. Indocyanine green (ICG) has been the most widely used fluorescent label, largely due to its availability as a licensed agent and its ease of application. The further development of alternative radiolabels or improved fluorescent tags will require investment in the development and licensing. There is also a vast potential for the targeting of albumin nanocolloids using existing strategies, which could be promising for the development of both diagnostic and therapeutic agents.

## 1. Background

### 1.1. Albumin Nanoparticle-Based Pharmaceuticals

Albumin nanoparticles are generally used as a drug delivery system for therapeutic agents, particularly chemotherapy [1]. The goal is to improve the delivery of the intact drug to the target and minimise the normal tissue exposure and damage, in particular bone marrow suppression [2,3]. A good example is nanoparticle albumin-bound paclitaxel (nab-PTX, Abraxane^®^), which can be administered with a shorter infusion time and less premedication, yet with efficacy better than soluble PTX. It was approved in 2005 for metastatic breast cancer [4], with indications later expanded to include non-small cell lung cancer and advanced pancreatic cancer. Nab-PTX particles are ~130 nm in diameter. Following intravenous administration, the nanoparticles dissociate rapidly into soluble, albumin-bound complexes of ~10 nm in size. These complexes undergo enhanced transendothelial transport, possibly mediated by the gp-60 albumin receptor, increasing the delivery of paclitaxel to the tumour [5]. Nab-PTX has been highly commercially successful, with worldwide sales exceeding USD 1.2 billion annually, though this has peaked due to the loss of market exclusivity with the licensing of generic equivalents [6].

### 1.2. Colloids as Radiopharmaceuticals

Radiopharmaceuticals are radioactive drugs administered to a patient, generally for diagnostic purposes. The behaviour of the radiopharmaceutical in the body is monitored externally with a gamma camera or scintillation probe. Diagnostic radiopharmaceuticals are given in subtherapeutic quantities and have an excellent safety profile. The most common radionuclide is the gamma emitter technetium-99m (^99m^Tc) which has a half-life of 6 h and is obtained from a generator on site.

Colloids have been used as radiopharmaceuticals for 75 years, beginning with the therapeutic application of inorganic gold colloid [7]. The first reports of ^131^I-labelled albumin colloid appeared in 1966 [8]. In terms of imaging, ^99m^Tc sulphide colloid has been used for liver/spleen imaging since the 1960s; however, this was an inorganic colloid [9]. It was joined by ^99m^Tc albumin nanocolloid in the 1980s [10].

Four routes of administration have been employed. These are by (1) intravenous injection, following which the radiocolloid localises due to phagocytosis in the reticuloendothelial system, allowing the imaging of the liver, spleen, and bone marrow; (2) interstitial injection, following which the radiocolloid is carried away by the lymphatic flow, outlining the lymphatic channels (lymphoscintigraphy) and allowing the identification of the sentinel lymph node; (3) intratumoural injection of therapeutic radiocolloids (not further discussed in this review); and (4) oral administration as a non-absorbable marker for gastric emptying studies (not further discussed in this review). Although the use of liver/spleen scanning has been in decline for decades due to the availability of alternative imaging modalities (e.g., computed tomography, ultrasound), the use of lymphoscintigraphy and, in particular, sentinel node detection has expanded greatly in the past 25 years.

The sentinel lymph node (SLN) is defined as the first node to which a cancer is most likely to have spread from a primary tumour. If the SLN can be identified and biopsied, a decision can be made on the extent of surgical resection which is required, with a goal of minimising surgery and post-surgical sequelae without adversely affecting the outcome. Although the concept was first proposed in 1960 [11], it did not come into widespread use until the 1990s [12]. Initially, a blue dye was injected in or around the tumour, and the SLN was identified visually during surgery. It quickly became apparent that a radiotracer would be a useful adjunct in the localisation of the SLN, and such agents are mostly radiocolloids. More recently, hybrid agents have been developed containing both radioactive and optical signals.

Radiopharmaceuticals for two modalities of nuclear imaging will be discussed. Gamma emitters such as ^99m^Tc are used for planar gamma imaging or single-photon emission computed tomography (SPECT), whereas positron emitters such as gallium-68 (^68^Ga) and zirconium-89 (^89^Zr) are used with positron-emission tomography (PET). The physical properties of these radionuclides are summarised in Table 1.

### 1.3. Objectives

For a general introduction to the challenges in the design of radiopharmaceuticals, there is an excellent recent review by the group from Leuven [13]. We will now discuss the specific design challenges in the preparation of albumin nanocolloid-based radiopharmaceuticals, including the physical properties (particle size and surface charge), targeting (passive or active), derivatisation for radiolabelling, derivatisation for dual-modality studies, and a demonstration that radiolabelling does not affect targeting. We will then look at the production challenges, including radiolabelling and purification, the maintenance of sterility and apyrogenicity, and quality assurance.

## 2. Design Challenges

### 2.1. Physical Characteristics: Particle Size and Surface Charge

The main physical characteristics which govern the biodistribution of nanoparticles are particle size and surface charge. Nanoparticles, by definition, have a diameter in the nanometre range, i.e., less than 1 micrometre.

In terms of interstitial injections for lymphatic studies, particle size plays a pivotal role in biological behaviour and hence the clinical application. Small colloidal particles (<5 nm in diameter) are liable to penetrate capillaries and be carried away by the circulating blood. At the opposite end of the spectrum, large colloidal particles (400–1000 nm) will not be carried away by the lymph flow and will remain in the injection depot. Within the useful range between these two limits, there is not absolute agreement on the optimal particle size. Some authors feel that the upper portion of the range (100–400 nm) is preferred because the particles should be trapped only in the first-echelon nodes, at the expense of slower and less extensive migration due to size. Others feel the lower portion of the range (5–100 nm) allows more rapid and extensive delivery, though at the expense of the possible overestimation of the number of sentinel nodes [14,15].

There are limitations in the variety of particle-sizing techniques which are used, including transmission electron microscopy (EM), scanning EM, light microscopy, filtration, centrifugation, Coulter counter, atomic force microscopy (AFM), photon correlation spectroscopy (PCS), and gel chromatography. These measure different parameters and are not directly comparable [16,17]. There can be a discrepancy between the physical measurements of the particle size and the measurements of the distribution of radioactivity; larger particles have a greater surface area, and thus, more atoms of ^99m^Tc can attach to a single particle [18]. Thus, radioactivity measurements using filtration techniques will give an apparently larger mean size, although this may actually be the most useful parameter in practice [18,19]. 

The ^99m^Tc-labelled human serum albumin nanocolloid (HSA-NC) was developed for inflammation imaging but quickly became used for lymphoscintigraphy [19]. HSA is heat denatured in the presence of stannous chloride, formulated with excipients such as a surface-active agent (e.g., poloxamer 238), then lyophilised, with a shelf-life of 1 year or more. It is stated that >95% of the particles are <80 nm in diameter, though measurements suggest a relatively narrow range of 7–15 nm [18,20]. The ^99m^Tc-HSA-NC complex forms when the lyophilised powder is reconstituted with ^99m^Tc-pertechnetate solution. In addition to the original product (Nanocoll^®^, Sorin/GE Healthcare, Chicago, IL, USA), there are now two generic products licensed in Europe with the same specifications (Nanotop^®^, Rotop, Dresden, German; Nanoscan, Medi-Radiopharma, Budapest, Hungary), one with slightly different specifications (Nanoalbumon^®^, Medi-Radiopharma) and another with a larger particle size range of 100–600 nm (Sentiscint^®^, Medi-Radiopharma). The characteristics of these agents are summarised in Table 2. HSA-NC is recommended in most of the European procedure guidelines for sentinel node localisation [21,22].

Negatively charged particles tend to be adsorbed on plasma proteins, whereas neutral or slightly positive charges undergo less adsorption [23]. Surface charge can be manipulated using a variety of constituents, but this does not appear to have been explored extensively with albumin nanoparticle radiopharmaceuticals. However, it remains an avenue for future research [2].

### 2.2. Targeting: Passive or Active

Until now, the targeting of HSA-NC radiopharmaceuticals has been passive and primarily determined by particle size.

For intravenously administered HSA-NC, the distribution amongst liver, spleen, and bone marrow is, in part, a function of the particle size. Particles < 100 nm tend to travel to the bone marrow, particles > 300 nm travel to the liver, and particles ~1000 nm are trapped in the spleen [24].

The fate of subcutaneously/interstitially administered radiocolloids has been outlined as the following sequence: phagocytosis by histocytes at the injection site; transport via lymph vessels to the regional lymph nodes; phagocytosis by macrophages in the lymph nodes; transport via blood vessels to the liver, spleen, and bone marrow; and elimination within the reticuloendothelial system [25].

Following interstitial administration, the targeting for lymph node specificity—side chains containing mannose which will bind to the CD206 mannose receptor on the macrophages in lymph nodes—is actually on a soluble molecule rather than an HSA-NC [26,27].

**Table 2 molecules-27-08596-t002:** Properties of licensed albumin nanocolloid radiopharmaceuticals.

Product (Manufacturer)	Specification or Measurement	Method *	References
Nanocoll (Sorin/GE Healthcare)	≥95% of particles < 80 nm in diameter	Not stated	Monograph
	>85% of particles 7–15 nm in diameter	PCS	[20]
	Mean diameter 12 nm	PCS	[20]
	Mean diameter 6.3 nm	PCS	[28]
	Mean diameter 8 nm	PCS	[29]
	Mean diameter 56 nm	ACS	[17]
	Mean diameter 9 nm	DLS	[18]
Nanotop (Rotop)	≥95% of particles ≤ 80 nm in diameter	Filtration	Monograph
	Mean diameter 7 nm	DLS	[18]
Nanoscan (Medi-Radiopharma)	≥95% of particles ≤ 80 nm in diameter	Not stated	Monograph
Nanoalbumon (Medi-Radiopharma)	>80% of particles < 100 nm in diameter	Not stated	Monograph
	Mean diameter 18 nm	DLS	[18]
SentiScint (Medi-Radiopharma)	>80% of particles 100–600 nm in diameter	PCS	Monograph

* PCS = photon correlation spectroscopy, ACS = autocorrelation spectroscopy, DLS = dynamic light scattering.

In addition to the phagocytosis-based localisation of HSA-NC, much of the localisation of albumin nanoparticle-based chemotherapy is attributed to enhanced permeability and retention (EPR), by which the higher permeability of the blood vessels in tumours allows the extravasation of the nanoparticles into the tumour microenvironment, where retention is further enhanced by the lack of efficient lymphatic drainage [30]. However, the EPR theory has recently been described as too simplistic, and it has been demonstrated that there can be the transport of nanoparticles across intact endothelium [31].

HSA-NC is highly amenable to modification for specific targeting [23,30]. The vast range of possibilities include folate [32], peptides (e.g., Arg-Gly-Asp RGD peptides which bind α_V_β_3_ integrin in neoangiogenesis) [33], and monoclonal antibodies (e.g., trastuzumab to target HER2-overexpressing breast tumour cells) [34]. Moreover, polyethylene glycol (PEG) can be attached to increase the residence time in the circulation [35]. This can increase the delivery to the target site, though this is not strictly targeting.

### 2.3. Derivatisation for Radiolabelling

Traditionally HSA-NC has been labelled with ^99m^Tc in a non-specific manner. The heat-denaturation of HSA in the presence of stannous chloride traps the stannous reducing agent within the particles. Upon the addition of ^99m^Tc-pertechnetate, reduction takes place, and the ^99m^Tc is chelated by sulphur and nitrogen atoms within the HSA primary structure [36]. Figure 1 outlines the preparation of HSA-NC, its radiolabelling with ^99m^Tc and the other radionuclides discussed subsequently, and the preparation of dual-modality agents via coupling with the fluorescent dye, indocyanine green (ICG).

There have been two recent reviews of radiolabelling approaches for nanoparticles, though these are not albumin-based [37,38].

Of more relevance to the current review, a group from Pavia, Italy have labelled commercially available HSA-NC (and macroaggregated albumin) with the positron emitter ^68^Ga as well as ^99m^Tc [39]. They incubated ^99m^Tc-HSA-NC or kit contents with ^68^Ga chloride in phosphate or acetate buffer at 75 °C for 15 min and obtained ~97% incorporation of the label. However, the ^68^Ga label was less stable than ^99m^Tc, with only 80% of the label retained after 1 h challenge with serum at 37 °C. Particle size did not appear to be changed appreciably. In subsequent work, the same group studied binding kinetics and capacity of HSA-NC for the two radionuclides. It appears that both bind to similar features of the albumin molecule, but the two do not directly compete. ^68^Ga seems to have a particular affinity for the free Cys in position 34 of the albumin chain [40]. It does not appear that any clinical results have been reported yet.

A slightly earlier study on the ^68^Ga labelling of nanoparticles for sentinel node PET imaging used a more traditional approach for the derivatisation of the nanoparticles with a chelator to bind ^68^Ga, although this study used iron oxide particles rather than albumin nanocolloid [41]. An amine moiety on the dextran coat of the iron oxide nanoparticles was conjugated with a DTPA chelator, which was then labelled with ^68^Ga. A pilot study was performed on five patients.

Another positron emitter, ^89^Zr, has also been used to label HSA-NC by a group from Amsterdam [42]. Once again, a commercial kit was used, but, in this case, a chelator was coupled to the HSA-NC, desferrioxamine B (DFO), via an isothiocyanatobenzyl linker at pH 9. The conjugate was then purified using size-exclusion chromatography and incubated with ^89^Zr oxalate for 1 h, followed by another purification. The labelling yield was 70–75%, and the radiochemical purity was >95% and stable for at least 24 h. Once again, the particle size did not appear to change appreciably. In a rabbit model, the biodistribution of the ^99m^Tc- and ^89^Zr-DFO-HSA-NC were remarkably similar. The feasibility of the approach was demonstrated in a pilot clinical study on patients with oral cancer [43,44].

### 2.4. Derivatisation for Dual-Modality Studies

In the last decade, there has been interest in hybrid tracers for multimodal imaging, specifically radioactive plus fluorescent. The first application of this to HSA-NC was the complexing of ^99m^Tc-HSA-NC with the fluorescent dye, indocyanine green (ICG) [45]. One of the drivers of this was purely practical: both HSA-NC and ICG were licensed drugs, thus could be used in patients without the requirement for separate licensing. As a result, it is the only such hybrid agent in routine clinical use [46].

The dual radiolabelled ^99m^Tc- and ^68^Ga-HSA-NC discussed in Section 2.3 above has also been complexed with ICG, forming a trimodal agent [39]. The state of the art has been recently reviewed by van Leeuwen, a pioneer in this area [46]. There are limitations to the use of ICG regarding its light output and its chemical stability over time.

The fluorescent dye IRDye 800CW has been covalently coupled to pharmaceutical grade HSA-NC without affecting the particle size [47]. This construct showed great stability over time. However, challenges remain in the incorporation of fluorescent tags into the radiopharmaceuticals, and this has also been reviewed by van Leeuwen [48]. For example, large fluorescent labels can alter the pharmacokinetics of HSA-NC. There can be problems with the chemical and radiochemical stability. The radiostability can be compromised by the photobleaching of the dye by free radicals produced by radioactive decay [49]. Finally, there is the high cost of licensing as a pharmaceutical, which is the reason for the continuing popularity of ICG [48].

### 2.5. Demonstration That Radiolabelling Does Not Affect Targeting

A new agent is commonly compared to a gold standard; however, for sentinel node imaging, the gold standard varies around the world, as has been summarised [19,25]. Despite the variety of agents with different properties used in different jurisdictions, there is no indication that there are geographical differences in the sensitivity of the sentinel node procedure [19].

Most of the examples cited above show a degree of validation. For example, here are three of the studies which demonstrated that ICG coupled ^99m^Tc-HSA-NC is not inferior to either component alone [45,50,51]. KleinJan et al. used in vitro techniques to mimic the clinical situation [52]. However, as most targeting of HSA-NC is passive, it can be difficult to validate; this is more of an issue with active targeting.

## 3. Preparation Challenges

### 3.1. Radiolabelling and Purification

As a general rule, the addition of the radioactive moiety is the final or penultimate step in the preparation of a radiopharmaceutical, as the clock begins ticking on the useful shelf-life of the agent once the radioactivity has been added. It is desirable that the efficiency of the radiolabelling step be as high as possible to minimise the cost and radiation exposure with a higher amount of starting radioactivity. For most PET radiopharmaceuticals, the radiolabelling is followed by purification and reformulation steps. With ^99m^Tc radiopharmaceuticals, it is usually simpler, involving the addition of ^99m^Tc-pertechnetate from an on-site generator to a “kit” which contains all of the non-radioactive components in quantities optimized to ensure a labelling efficiency of >95%. Furthermore, the kits are manufactured to pharmaceutical standards.

When the radiocolloid is required, a freeze-dried HSA-NC kit is reconstituted with ^99m^Tc pertechnetate, the ^99m^Tc-HSA-NC complex forms rapidly (within s to min) [40] and remains stable for many hours. No purification step is required. The labelling efficiency can be checked using thin layer chromatography and the maximum particle size using membrane filtration (see below). Although the manufacturer generally specifies an expiry time of 6 h after preparation, stability up to 30 h has been demonstrated [28].

Though the physical steps in the routine preparation of ^99m^Tc-HSA-NC may appear as simple as the reconstitution of a freeze-dried powder, the reality is much more complex. On each occasion, a set of chemical reactions are initiated which are subject to potential problems. The ^99m^Tc pertechnetate is reduced using stannous chloride in an anoxic environment. The reduced ^99m^Tc then binds to the functional groups on the denatured HSA. The entry of oxygen can result in the oxidation of ^99m^Tc back to free ^99m^Tc pertechnetate, which is soluble and would rapidly diffuse away from the injection depot. Other side reactions can lead to ^99m^Tc colloids which might have different properties from the desired complex. If the amount of ^99m^Tc activity added is too high, radiolysis of the desired complex can occur. With optimised formulations, product quality is very reliable, and routine radiochemical purity testing only checks for the amount of free ^99m^Tc pertechnetate.

For the preparation of the fluorescent ICG complex mentioned above, ^99m^Tc-HSA-NC is prepared in the usual manner, incubated for 30 min, and then a small quantity (0.25 mg) of pharmaceutical-grade ICG is added; the complex forms virtually instantaneously [53]. Again, no purification is required.

For the preparation of the dual-labelled ^68^Ga/^99m^Tc-HSA-NC, the ^99m^Tc-HSA-NC is prepared as usual, then incubated at 75 °C for 15 min with ^68^Ga obtained from an on-site generator, and the product is ready to use [39,40]. The ^68^Ga labelling step can be performed using an automated synthesis module to reduce the occupational radiation exposure and ensure aseptic conditions (see below).

Finally, ^89^Zr-DFO-HSA-NC can be prepared using a licensed HSA-NC kit via several steps. The HSA-NC is first coupled with desferrioxamine B and purified using size-exclusion chromatography. The DFO-HSA-NC is then incubated with ^89^Zr oxalate and again purified to yield a product with >95% radiochemical purity. The final product is sterilised by its passage through a 220 nm membrane filter [42,43,44].

### 3.2. Maintenance of Sterility and Apyrogenicity

Radiopharmaceuticals have short half-lives (min to h), meaning there is insufficient time for sterility testing before release; indeed, radiopharmaceuticals may be the only class of injectable drugs released without sterility testing having been performed. Therefore, all the components must be certified sterile, and the assembly must be performed under aseptic conditions with environmental monitoring. While, for many years, this meant only the use of sterile gloves and wiping the septum of the vial with an alcohol swab, in many countries, special facilities with a Grade A or Class 100 environment are now required [54]. With albumin products, there is not the possibility of post-labelling heat sterilisation, even if the radioactive half-life allowed it.

After the facilities and environment, the second concern is the pharmaceutical quality of the ingredients. In the case of HSA-NC, this means the source and quality of albumin. Some manufacturers of licensed HSA-NC kits state that the albumin is derived from human blood donations tested according to European regulations and found non-reactive for the hepatitis B surface antigen, antibodies to the human immunodeficiency virus, and antibodies to the hepatitis C virus. Olsen has reported that there can be problems with pharmaceutical grade HSA [55]. Another possibility would be recombinant human albumin [56], though the only biological radiopharmaceutical studied with rhHSA is macroaggregated albumin [57].

Most of the examples cited above have avoided the dilemma of albumin quality by using a licensed HSA-NC kit. Thus, the preparation of ^99m^Tc-HSA-NC and even ^68^Ga/^99m^Tc-HSA-NC is essentially a standard procedure, as is the addition of ICG to the radioactive complex. However, in the case of ^89^Zr-HSA-NC, considerable manipulation of the product is required, including two purification steps on size-exclusion columns. Although sterile columns are commercially available, the procedure is too cumbersome for routine use. If the ^89^Zr-labelled agent is to become commercially viable, there will need to be an improved formulation with fewer interventions. Certainly, the intermediate DFO-HSA-NC could be produced at a pharmaceutical grade, and perhaps, the radiolabelling conditions could be optimised or a more efficient chelator chosen to obviate the need for post-labelling purification. Again, we can see why ICG-^99m^Tc-HSA-NC is the only product in widespread clinical use at present.

### 3.3. Quality Assurance

After a radiopharmaceutical has been prepared, it must undergo a set of predetermined quality control procedures before it is released for use. At a minimum, these usually include total radioactivity, volume, and appearance but may also include radiochemical purity (RCP), radionuclidic purity, chemical purity, pH, particle size, endotoxin content, and osmolality [58]. Some tests can be performed after release, such as sterility testing (because it takes too long) and some chemical tests (which can only be performed after radioactive decay). In these cases, there must be satisfactory results from a series of validation batches.

There is disagreement about the need for the RCP testing of licensed products, such as ^99m^Tc-HSA-NC [59]. Thin-layer chromatography is commonly used [18,60]. For non-licensed products, including those prepared outside the manufacturer’s directions for a licensed product, RCP must be tested prior to release. Convenient methods have been reported for the analysis of ^68^Ga/^99m^Tc-HSA-NC [39] and ^89^Zr-DFO-HSA-NC [42].

The determination of the extent of the binding of ICG to HSA-NC can be a more difficult proposition (JR Ballinger, unpublished results); however, Persico et al. report a ≥98% binding of ICG as determined using thin-layer chromatography followed by imaging with a handheld fluorescence probe [39]. KleinJan et al. separated the components using size-exclusion column chromatography followed by UV-Vis spectrophotometry [52].

## 4. Conclusions

The primary use of albumin nanoparticle-based radiopharmaceuticals is in sentinel node localisation, a widely adopted procedure as an adjunct to surgery in a wide variety of solid tumours. There are several essentially equivalent licensed HSA-NC products available, at least in Europe.

Dual- (or triple-) modality detection has become a reality, with virtually all clinical work being carried out with ICG as the fluorescent marker due to its availability as a licensed agent and the simplicity of its application. However, with sufficient commercial interest, superior agents could be introduced [46].

There is vast scope for the targeting of HSA-NC using approaches which have been shown to be useful with other agents. This could be very promising for both diagnostic and therapeutic agents [1,2].

## Figures and Tables

**Figure 1 molecules-27-08596-f001:**
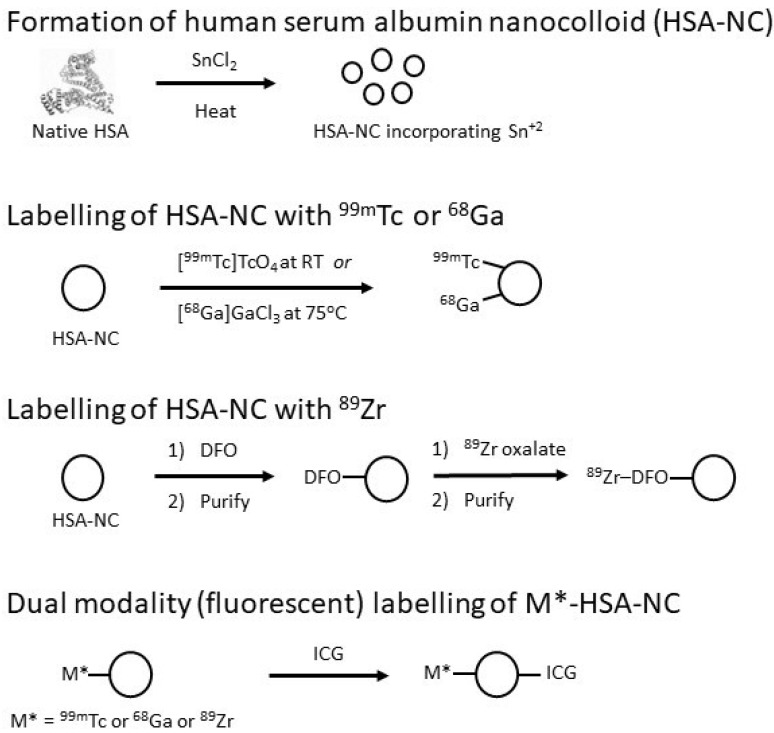
Graphical representation of the sequence of steps in preparation and radiolabelling of HAS-NC and its dual-modality analogue.

**Table 1 molecules-27-08596-t001:** Physical properties of radionuclides discussed.

Radionuclide	Half-Life	Emission (Abundance), Mean Energy	Route of Production
Technetium-99m, ^99m^Tc	6 h	Gamma (89%), 140 keV	Generator (or cyclotron)
Gallium-68, ^68^Ga	1.1 h	Positron (89%), 0.89 MeV	Generator or cyclotron
Zirconium-89, ^89^Zr	78 h	Positron (23%), 0.39 MeV	Cyclotron

## Data Availability

Not applicable.

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
