# Peer review of "Challenges in Preparation of Albumin Nanoparticle-Based Radiopharmaceuticals"

_molecules, 2022, doi:10.3390/molecules27238596_

Round 1

Reviewer 1 Report

The topic is of interest and the idea is worthy. The review design is well organized, and ideas are clearly presented. Over the manuscript is very well articulated and has very convincing approach.The article requires some minor revisions:

1-The author should add one or two schematic diagrams to explain the idea of Design and formulation challenges.

2-The authors may add a new paragraph to the targeting section to discuss the different types of nano drug delivery systems and their medical application such as improving the bioavailability of drugs also its role for treatment of cancers. Please see these references:

https://www.sciencedirect.com/science/article/abs/pii/S1773224721000010

https://pubmed.ncbi.nlm.nih.gov/33781877/

Reviewer 2 Report

The purpose of this review is to increase the reader’s knowledge and understanding of problems associated with the preparation and dispensing of albumin nanoparticle-based radiopharmaceuticals and the corresponding effects/manifestations of these problems. The widespread reporting of such problems in a timely manner will contribute to improved safety and efficacy of radiopharmaceutical. Many of these problems have been described in past reviews on this topic (Brandt M., Cardinale J., Giammei C., Guarrochena X. Happl B., Jouini N., Mindt T.L. Mini-review: Targeted radiopharmaceuticals incorporating reversible, low molecular weight albumin binders. Nucl. Med. Biol. 2019, 70, 46–52. doi: 10.1016/j.nucmedbio.2019.01.006. Lau J., Jacobson O., Niu G., Lin K. S., Benard F., Chen X. Bench to Bedside: Albumin Binders for Improved Cancer Radioligand TherapiesBioconjugate Chem. 201930487– 502 doi: 10.1021/acs.bioconjchem.8b00919.  Vermeulen K., Vandamme M., Bormans G., Cleeren F. Design and Challenges of Radiopharmaceuticals. Seminars in Nuclear Medicine. 2019, 49, 339–356. doi.org/10.1053/j.semnuclmed.2019.07.001).

I find the novelty of the manuscript to be limited as it is a variation on a review that was published earlier (Ballinger, J. The Use of Protein Based Radiocolloids for Sentinel Node Localisation. Clin Transl Imaging 2015, 3, 179-186).  It is not clear how this review differs from the prior reviews reported by the author (Ballinger, J. 2015; Ballinger, J. and Blower P.J., 2011).

Introduction is not in line with the discussed content. The introductory section is well written in terms of language. However, it is out of focus and does not match the content of the remainder of the review, nor is it aligned with the goals that are put forward.

The chapter 2.2 aims to summarize the results regarding non radiolabeled serum albumin-based nanoparticles for drug delivery purposes. In particular, it focuses on the relationship between their preparation techniques and passive or active targeting. In contrast to conventional albumin nanoparticle-based pharmaceuticals, radiopharmaceuticals have several unique characteristics that are potentially problematic in their preparation and dispensing: 1. Their preparation involves chemical reactions that may produce undesired radiochemical impurities; 2. Their emitted radiation, especially at high intensities, may produce radiolytic effects that can result in undesired impurities; 3. Their chemical properties, especially in combination with their small mass quantities, may result in undesired adsorption to container components or interaction with trace contaminants leached therefrom. These problems may subsequently result in unexpected alterations in biodistribution and/or inadequate localization in organs of interest, and thereby interfere with diagnostic interpretation. However, a meaningful discussion of these challenges in preparation of albumin nanoparticle-based radiopharmaceuticals is lacking.

While the information and the accompanying discussions in the chapters 2.3 and 2.4 are certainly interesting, they do not add sufficient weight to justify publication as such in Molecules.
